# Numbers as Text: Complementary Dual-Modality Embeddings for Time Series Forecasting

## Abstract

The remarkable success of Large Language Models (LLMs) in language tasks inspires us to explore their application in long-term time series forecasting(LTSF). Their ability to capture complex sequence dependencies from massive datasets suggests a strong potential for modeling the intricate patterns inherent in time series data. However, current methods for applying LLMs to LTSF often rely on hand-engineered statistical features and elaborate, dataset-specific prompts. This approach not only deviates from the end-to-end learning paradigm but also introduces a critical risk of lookahead bias, where performance gains may stem from the model accessing information within its pre-training corpus rather than genuine forecasting ability. A clear gap exists for a method that leverages LLMs on raw time series data in a robust, feature-free manner. To address this gap, we propose a novel framework, NumText, that directly operates on raw time series data. Our method treats the series as a dual-modality input, generating two parallel representations: a direct numerical value embedding and a forecasting-oriented LLM embedding derived from the series' textual form. These distinct embeddings are then combined through a modality-specific Mixture-of-Experts (MoE) to form a rich, unified input for a downstream attention mechanism. Furthermore, we introduce a time-series text embedding cache to reduce computational overhead during inference. Our extensive experiments reveal that the numerical value embeddings and the LLM's textual embeddings are highly complementary, capturing different yet synergistic signals crucial for forecasting. This synergy enables our model to achieve improvement upon current state-of-the-art (SOTA) performance on several benchmark TSF datasets, establishing a more robust approach for applying LLMs in this domain.

## 1 INTRODUCTION

Long-term time series forecasting (Li et al., 2023) is a critical task in numerous domains, from financial markets (Cao et al., 2019) to energy consumption (Wang et al., 2020) and climate modeling (Guo et al., 2024). Recently, the remarkable capabilities of LLMs Patil & Gudivada (2024) in natural language processing (NLP) have inspired a wave of research into their application for LTSF. While some studies have shown promising results (Xue & Salim, 2023; Jin et al., 2024; Liu et al., 2025), their methodologies raise critical questions. To date, most approaches rely on deliberately designed prompts that incorporate hand-engineered statistical features and dataset-specific metadata. This dependency on auxiliary information creates two fundamental problems: (1) it departs from the end-to-end learning paradigm, making the model reliant on manual feature crafting; and (2) it introduces a significant risk of lookahead bias, where performance gains may be attributable to the LLM accessing knowledge about future data from its vast pre-training corpus, rather than genuine forecasting ability.

An intuitive solution is to discard these prompts and apply LLMs directly to raw time series data, like the 3rd format in Figure 1. However, this intuitive approach remains largely unexplored in published research, indicating its performance may be suboptimal. We hypothesize that the primary challenge is developing an effective input representation for raw time series data. Simply inputting a time series as a sequence of float numbers is insufficient for pre-trained LLMs. In fact, recent work

suggests that LLMs possess a "fragile number sense," where their ability to reason with numbers is based more on pattern matching learned algorithms than a genuine comprehension of numerical concepts like magnitude or place value (Rahman & Mishra, 2025). Concurrently, the broader time series forecasting field has also seen a shift towards innovative input representations. Techniques such as patching, which segments a series into smaller windows, have emerged as a powerful method to preserve local information and improve computational efficiency (Nie et al., 2023). Another approach is quantization, which discretizes numbers into a word-like format that LLMs can process(Ansari et al., 2024). We hypothesize that such structural priors are a crucial, yet missing, ingredient for successfully applying LLMs to raw time series.

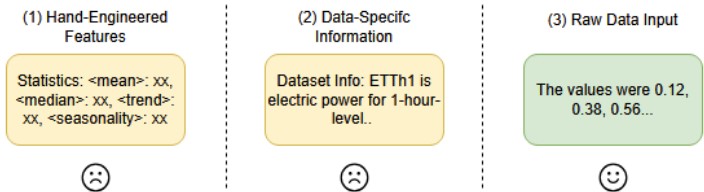

Figure 1: Three paradigms for encoding time series data as input for LLMs. (1) Using hand-engineered statistical features. (2) Using dataset-specific metadata as a prompt. (3) Using the raw numerical sequence directly. Our work investigates improvements to the raw data paradigm (3).

In this paper, we argue that the ideal representation is neither purely numerical nor purely textual, but a synergistic combination of both. We introduce the NumText framework, a novel dual-modality approach that learns from two complementary views of the same raw time series. Our core hypothesis is that LLMs, through their vast pre-training on sequential data, have developed latent sequence modeling and forecasting abilities. A direct numerical embedding, while precise, lacks these powerful learned priors. By representing the time series as text, we can unlock the LLM's intrinsic forecasting capabilities. We therefore combine two complementary views: for the numerical form, we create a direct numerical value embedding that preserves precise magnitudes and local patterns, and for the textual form, we treat the patched values as text tokens to derive an LLM embedding, which captures rich semantic and contextual relationships. These parallel embeddings are then combined through a modality-specific Mixture-of-Experts and fed into a downstream forecasting model. To enhance efficiency, we also introduce a time-series text embedding cache.

Our experiments validate this dual-modality hypothesis, showing that the two representations provide complementary signals essential for robust forecasting. By unifying them, our model achieves improvement over current SOTA results across several benchmark LTSF datasets. This work establishes a more methodologically sound and performant approach for leveraging LLMs in this domain. Our contributions are threefold:

- We propose NumText that, for the first time, successfully applies LLMs to raw time series data in a truly end-to-end fashion, eliminating the risk of data leakage and lookahead bias inherent in prompt-based methods, creating a new benchmark.

- We introduce a dual-modality architecture and are the first to empirically demonstrate that direct numerical embeddings and LLM-derived textual embeddings of time series are highly complementary, using an MoE block to combine, providing a crucial insight into time series representation learning.

- NumText achieves state-of-the-art performance on multiple benchmark datasets, validating our approach and establishing the multi-modal representation of unimodal data as a powerful and promising new direction for the time series forecasting field.

## 2 RELATED WORKS

**LLM for Time Series Forecasting.** The application of Large Language Models (LLMs) to time series forecasting has recently gained significant traction. A dominant paradigm uses LLMs as forecasters guided by elaborate prompts containing hand-engineered statistical features and dataset

metadata (Gruver et al., 2024; Jin et al., 2024; Liu et al., 2025; Cao et al., 2024). Among them, **TimeCMA** (Liu et al., 2025) achieves SOTA LTSF performance. While some methods explore inputting numerical embeddings directly into an LLM (Zhou et al., 2023), others have questioned the overall utility of the LLM component, suggesting simpler architectures may be sufficient (Tan et al., 2024). Our work addresses a critical gap in this landscape. Instead of relying on hand-engineered features, and auxiliary metadata, we are the first to systematically demonstrate that a textual representation of raw time series numbers, when combined with a direct numerical representation, provides a more robust and methodologically sound approach for leveraging LLMs.

**Time Series Representation Learning.** The focus in time series forecasting has increasingly shifted from architectural innovation to representation learning. A pivotal moment was the introduction of Dlinear (Zeng et al., 2023), a simple linear model that challenged the prevailing effectiveness of complex Transformer-based architectures and highlighted the importance of the input representation. Patching, as popularized by PatchTST (Nie et al., 2023), has emerged as a powerful technique for creating locally-aware, subseries-level tokens. Other innovations include iTransformer's approach of inverting the temporal and variable dimensions to create variate-level tokens (Liu et al., 2024), and Chrono's idea of quantization, converting real values into discrete tokens (Ansari et al., 2024). Our work builds upon these advances, adopting patching and quantization as foundational pre-processing steps.

**Multi-Modality Fusion.** As models incorporate diverse data sources, effective fusion mechanisms have become crucial. Common strategies range from simple concatenation (Jin et al., 2024) to more complex attention-based methods like self-attention and cross-attention (Zhong et al., 2025). Our approach utilizes a modality-specific Mixture-of-Experts (MoE) network (Shazeer et al., 2017). Unlike static fusion methods, MoE provides a dynamic, learnable mechanism that weights the contribution of each modality—numerical and textual—based on the input data, allowing for a more adaptive and context-aware representation.

**Forecasting Backbones.** While representation is critical, the choice of a forecasting backbone remains important for modeling temporal dependencies. Transformer-based architectures (Vaswani et al., 2017) are particularly well-suited for this task due to their ability to capture long-range dependencies. Models such as Autoformer, Informer, and PatchTST have demonstrated state-of-the-art performance by leveraging sophisticated attention mechanisms to model relationships across extensive lookback windows (Kitaev et al., 2020; Wu et al., 2021; Zhou et al., 2021; Liu et al., 2022; Zhang & Yan, 2023). In our NumText framework, we utilize a simple Transformer backbone to process the fused multi-modal representations. This choice allows the model to effectively learn from the rich signals provided by both the numerical and textual modalities, demonstrating that a powerful representation can enhance the performance of even standard backbone architectures.

## 3 METHODOLOGY

Figure 2 illustrates the overall architecture of NumText. The core of our approach is a dual-modality embedding strategy that processes each time series patch through two parallel, specialized experts: a *Textual Expert* that leverages a pre-trained LLM for semantic feature extraction, and a *Numerical Expert* that preserves the raw value information. A soft Mixture-of-Experts (MoE) module dynamically fuses the outputs of these experts into a unified, rich representation. This representation is then fed into a standard Transformer encoder and a forecasting head to generate the final prediction.

### 3.1 NORMALIZATION & PATCHIFY

Let an input univariate time series be denoted as $X_{raw} \in \mathbb{R}^{L_{in}}$, where $L_{in}$ is the lookback window length. We first apply instance normalization to standardize the series. The mean $\mu$ and standard deviation $\sigma$ are computed over the lookback window, and the normalized series is $X = (X_{raw} - \mu)/\sigma$. This step makes the model invariant to shifts and scaling in the input data.

Following (Nie et al., 2023), we segment the normalized series $X$ into $N$ overlapping patches of length $P$ with a stride $S$. This results in a sequence of patches $X_p = \{x_1, x_2, \ldots, x_N\}$, where each patch $x_i \in \mathbb{R}^P$. The number of patches is $N = \lfloor (L_{in} - P)/S \rfloor + 1$. This patching technique effectively captures local temporal patterns and reduces the input sequence length for the subsequent attention mechanism, enhancing both efficiency and performance.

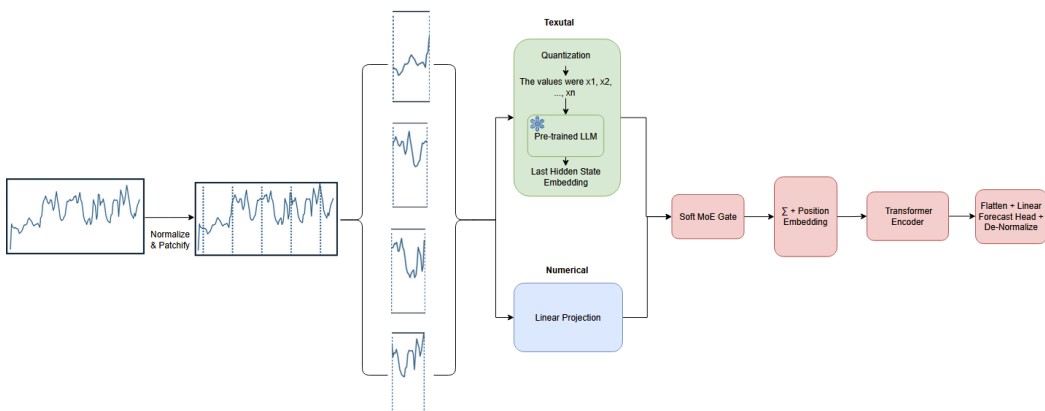

Figure 2: The overall architecture of NumText. An input time series is normalized and patched, then processed in parallel by a Textual Expert (using a frozen LLM) and a Numerical Expert (using a linear projection). A soft MoE module dynamically fuses their embeddings, which are then fed into a Transformer encoder to produce the forecast.

## 3.2 SOFT MIXTURE OF EXPERTS

To effectively integrate the complementary representations from our dual-modality experts, we employ a soft Mixture-of-Experts (MoE) framework. This module learns to dynamically weight the contribution of each expert for every input patch, creating a context-aware fused embedding.

A lightweight gating network, implemented as a two-layer MLP with a ReLU activation, determines the expert weights. The network takes the mean of all patches from the input series, $\bar{x} = \frac{1}{N} \sum_{i=1}^{N} x_i$, as a global context vector. This vector is passed through the gating network to produce a logit vector $g \in \mathbb{R}^2$. A Softmax function then normalizes these logits into weights, $[w_{num}, w_{text}] = \text{softmax}(g)$, ensuring they sum to one.

Given the numerical embedding $E_{num}^{(i)}$ and textual embedding $E_{text}^{(i)}$ for the $i$-th patch (described in Sections 3.3 and 3.4), the final fused embedding $E_{fused}^{(i)}$ is computed as their weighted sum:

$$E_{fused}^{(i)} = w_{num} \cdot E_{num}^{(i)} + w_{text} \cdot E_{text}^{(i)} \tag{1}$$

This soft combination allows the model to flexibly prioritize either the precise numerical values or the abstract semantic features, depending on which is more informative for the given time series.

## 3.3 TEXTUAL EXPERT

The Textual Expert is designed to harness the powerful sequence modeling and pattern recognition capabilities of a pre-trained LLM. For each patch $x_i \in \mathbb{R}^P$, we first transform its numerical values into a textual format that the LLM can process.

To bridge the gap between continuous time series values and the discrete token vocabulary of LLMs, we quantize the normalized values within each patch. This is motivated by the documented challenges LLMs face in processing floating-point numbers directly (Rahman & Mishra, 2025). We adopt an affine transformation similar to Ansari et al. (2024). The normalized values $v \in x_i$ are first quantized into a vocabulary of positive integers using a simple affine transformation: $v_{quant} = \text{round}((v + \alpha) \times \beta)$, where we use $\alpha = 3$ and $\beta = 100$ in our experiments. These quantized values are then formatted into a structured text prompt, "[v_1, v_2, ..., v_p].". Figure 3 shows an example of quantization. This process is predicated on the hypothesis that an LLM's forecasting contribution stems from recognizing high-level patterns and semantics, rather than interpreting precise numerical magnitudes(Garg et al., 2022).

After tokenization, this prompt is fed into a frozen pre-trained LLM (we use Llama-3.2-1B-Instruct). To obtain a fixed-size representation for the patch, we extract the final hidden states of the LLM

corresponding to the input tokens. We apply mean pooling over these hidden states to produce a single vector, $h_i \in \mathbb{R}^{d_{llm}}$. This vector is then projected to the model's hidden dimension, $d_{model}$, via a linear layer: $E_{text}^{(i)} = W_{text}h_i + b_{text}$.

To address the high computational cost of LLM inference, we implement a persistent caching strategy. The embedding for each unique text prompt is computed once and stored on disk. In subsequent training epochs or during inference, if the same prompt is encountered, its pre-computed embedding is retrieved from the cache, dramatically reducing latency.

### 3.4 NUMERICAL EXPERT

In contrast to the abstract representation from the Textual Expert, the Numerical Expert provides a direct, value-based embedding of each patch. This expert ensures that the precise magnitude and local temporal structure of the raw series are preserved.

The implementation is a straightforward linear projection. The normalized patch vector $x_i \in \mathbb{R}^P$ is mapped directly to the model's embedding dimension $d_{model}$ using a trainable weight matrix $W_{num}$ and bias $b_{num}$:

$$E_{num}^{(i)} = W_{num}x_i + b_{num} \quad (2)$$

This simple yet effective approach provides a clean numerical signal that complements the high-level semantic features captured by the LLM.

Figure 3: Illustration of our quantization process, where normalized floating-point values from a patch are converted into a discrete vocabulary of integers before being formatted as a text prompt for the LLM.

### 3.5 FORECASTING WITH TRANSFORMER BACKBONE

Once the sequence of fused patch embeddings, denoted as $E_{\text{fused}} = \{E_{\text{fused}}^{(1)}, \ldots, E_{\text{fused}}^{(N)}\}$, is obtained, it is prepared for the forecasting backbone. A fixed **Positional Encoding** ($PE$) is added to provide the model with information about the relative order of the patches. The values of the $PE \in \mathbb{R}^{N \times d_{\text{model}}}$ vector are determined by sine and cosine functions of different frequencies:

$$PE_{(pos,2i)} = \sin(pos/10000^{2i/d_{\text{model}}}) \quad (3)$$

$$PE_{(pos,2i+1)} = \cos(pos/10000^{2i/d_{\text{model}}}) \quad (4)$$

where $pos$ is the position of the patch in the sequence ($0 \leq pos < N$), $i$ is the dimension index ($0 \leq i < d_{\text{model}}/2$), and $d_{\text{model}}$ is the model's hidden dimension. The final input to the Transformer encoder, $X_{\text{enc}}$, is the sum of the fused embeddings and these positional encodings:

$$X_{\text{enc}} = E_{\text{fused}} + PE \quad (5)$$

The resulting sequence is then processed by a multi-layer **Transformer Encoder** (Vaswani et al., 2017). Each layer in the encoder performs two main operations:

- **Multi-Head Self-Attention:** This mechanism allows every patch to look at all other patches in the sequence and calculate scores to determine which ones are most relevant to its own context. The "multi-head" design enables the model to do this from multiple perspectives in parallel, capturing different types of temporal relationships (e.g., long-term trends and short-term seasonalities) simultaneously. The output is a refined representation for each patch, enriched with information from the entire sequence.

- **Feed-Forward Network (FFN):** Following the attention step, the representation of each patch is independently processed by a simple two-layer neural network. This step adds modeling capacity and allows for more complex patterns to be learned from the attention output.

Residual connections and layer normalization are applied around each of these two steps to ensure stable training and effective information flow through the deep network.

Finally, the output from the last encoder layer, a sequence of contextualized representations, is passed to a **Forecasting Head**. The sequence is flattened into a single vector and projected by a linear layer to produce a forecast for the desired prediction length, $L_{\text{out}}$. This prediction is denormalized using the saved mean and standard deviation to produce the final forecast in the original data scale.

# 4 THEORETICAL JUSTIFICATION OF MoE FOR DUAL-MODALITY FUSION

In this section, we provide a theoretical justification for employing a Mixture-of-Experts (MoE) to dynamically fuse two complementary modalities: numerical embeddings $E_{\text{num}}$ and textual embeddings $E_{\text{text}}$.

Let $Y$ denote the target, and define the Bayes optimal predictors Barbieri & Berger (2004)

$$\eta_{\text{num}}(x_{\text{num}}) = \mathbb{E}[Y \mid E_{\text{num}} = x_{\text{num}}], \qquad \eta_{\text{joint}}(x_{\text{num}}, x_{\text{text}}) = \mathbb{E}[Y \mid E_{\text{num}} = x_{\text{num}}, E_{\text{text}} = x_{\text{text}}].$$

**Theorem 4.1.** *The minimal mean squared error (MSE) using both modalities is never worse than using a single modality:*

$$\mathcal{R}(E_{num}, E_{text}) \leq \mathcal{R}(E_{num}),$$

*with strict inequality whenever $E_{text}$ provides non-redundant information about $Y$ given $E_{num}$.*

This theorem establishes that leveraging both modalities is theoretically guaranteed to reduce (or at least not increase) the Bayes risk.

Consider a linear fusion

$$f_w(x) = w_{\text{num}} E_{\text{num}}(x) + w_{\text{text}} E_{\text{text}}(x), \qquad w_{\text{num}} + w_{\text{text}} = 1.$$

**Theorem 4.2.** *Suppose the input distribution is a mixture $p(x) = \sum_{k=1}^{K} \pi_k p_k(x)$, and the optimal fusion weights differ across components, i.e. $w_k^* \neq w_\ell^*$ for some $k \neq \ell$. Then any fixed weight $w$ incurs strictly higher overall risk than an oracle dynamic scheme that selects $w_k^*$ for each component:*

$$\mathcal{R}_{static}(w) > \mathcal{R}_{dyn}^{oracle}.$$

This result justifies the use of MoE gating: by learning input-dependent weights $w(x)$, the model approaches the oracle dynamic fusion and thus achieves lower expected risk than any static fusion rule.

**Theorem 4.3.** *Assume the two experts produce unbiased predictions with conditional error vector*

$$\varepsilon(x) = \begin{bmatrix} \hat{y}_{num}(x) - \eta_{joint}(x) \\ \hat{y}_{text}(x) - \eta_{joint}(x) \end{bmatrix}, \quad \mathbb{E}[\varepsilon(x) \mid x] = 0,$$

*and conditional covariance $\Sigma(x) = \mathbb{E}[\varepsilon(x)\varepsilon(x)^\top \mid x]$. Then the conditional MSE of a convex combination $w(x) = (w_{num}(x), w_{text}(x))$ is*

$$\text{MSE}(x; w) = w(x)^\top \Sigma(x)\, w(x).$$

*The pointwise optimal weight is*

$$w^*(x) = \arg\min_{w:\, w_1 + w_2 = 1} w^\top \Sigma(x) w.$$

*If $\Sigma(x)$ varies with $x$ on a set of positive measure, then any fixed static weight $w$ has strictly larger overall expected risk than the dynamic choice $w^*(x)$.*

This theorem shows that when the error covariance of the two modalities depends on the input, dynamic gating adaptively selects the variance-minimizing weights and strictly improves over any fixed static fusion.

Table 1: Dataset statistics are from (Wu et al., 2023). The dataset size is organized in (training, validation, testing).

| Dataset | Series Length | Dataset Size | Frequency | Domain |
|---|---|---|---|---|
| ETTm1 | {96, 192, 336, 720} | (34465, 11521, 11521) | 15 min | Temperature |
| ETTm2 | {96, 192, 336, 720} | (34465, 11521, 11521) | 15 min | Temperature |
| ETTh1 | {96, 192, 336, 720} | (8545, 2881, 2881) | 1 hour | Temperature |
| ETTh2 | {96, 192, 336, 720} | (8545, 2881, 2881) | 1 hour | Temperature |
| Electricity | {96, 192, 336, 720} | (18317, 2633, 5261) | 1 hour | Electricity |
| Traffic | {96, 192, 336, 720} | (12185, 1757, 3509) | 1 hour | Transportation |
| Weather | {96, 192, 336, 720} | (36792, 5271, 10540) | 10 min | Weather |
| ILI | {24, 36, 48, 60} | (617, 74, 170) | 1 week | Illness |

Table 2: Comparison of model performance (MSE and MAE) across different modalities. Performances for Illness is averaged across {24, 36, 48, 60}, while the others are averaged across {96, 192, 336, 720} pred lengths. **Bold** indicates the best performance for each metric. Full detailed results can be found in Appendix A.4

| Data | NumText | | Num Only | | Text Only | |
|---|---|---|---|---|---|---|
| | MSE | MAE | MSE | MAE | MSE | MAE |
| ETTh1 | **0.079** | **0.217** | 0.081 | 0.220 | 0.082 | 0.225 |
| ETTh2 | **0.204** | **0.356** | 0.206 | 0.358 | 0.211 | 0.363 |
| ETTm1 | **0.052** | **0.172** | 0.053 | 0.172 | 0.053 | 0.173 |
| ETTm2 | 0.125 | 0.263 | **0.121** | **0.258** | 0.128 | 0.267 |
| Electricity | 0.406 | 0.470 | **0.395** | **0.458** | 0.427 | 0.484 |
| Traffic | **0.325** | **0.417** | 0.328 | 0.420 | 0.334 | 0.425 |
| Weather | **0.002** | **0.030** | 0.002 | 0.031 | **0.002** | **0.030** |
| Illness | **1.363** | **0.982** | 1.636 | 1.092 | 1.500 | 1.053 |

## 5 MAIN RESULTS

### 5.1 EXPERIMENTAL SETUP

We mainly follow the experimental configurations in (Wu et al., 2023) across all baselines within a unified evaluation pipeline in https://github.com/thuml/Time-Series-Library?tab=readme-ov-file for fair comparisons. We use Llama-3.2-1B-Instruct as the default backbone model unless stated otherwise. Our model implementation is on PyTorch.

We evaluate all models using Mean Squared Error (MSE) and Mean Absolute Error (MAE). Following the benchmark protocol, we forecast horizons of {96, 192, 336, 720} for most datasets, and {24, 36, 48, 60} for the Illness dataset.

### 5.2 DATASET

All our datasets and dataset statistics are summarized in Table 1. We evaluate the univariate long-term forecasting performance on the well-established eight different benchmarks.

### 5.3 PERFORMANCE BETWEEN MODALITIES

Table 2 presents our ablation study comparing the full **NumText** model against its unimodal components: **Num Only** (equivalent to PatchTST) and **Text Only**. The results clearly show that the dual-modality NumText is the top-performing model, achieving the best MSE and MAE on six out of the eight datasets.

Table 3: Comprehensive model performance comparison (MSE and MAE) across all datasets. **Bold** indicates the best performance for each metric. Full detailed results can be found in Appendix A.6

| Data | NumText | | PatchTST | | TimeCMA | | Autoformer | | DLinear | | iTransformer | | Crossformer | | Pyraformer | | Reformer | |
|---|---|---|---|---|---|---|---|---|---|---|---|---|---|---|---|---|---|---|
| | MSE | MAE | MSE | MAE | MSE | MAE | MSE | MAE | MSE | MAE | MSE | MAE | MSE | MAE | MSE | MAE | MSE | MAE |
| ETTh1 | **0.079** | **0.217** | 0.081 | 0.220 | 0.090 | 0.234 | 0.102 | 0.249 | 0.118 | 0.263 | 0.136 | 0.285 | 0.311 | 0.480 | 0.397 | 0.547 | 1.332 | 1.069 |
| ETTh2 | **0.204** | **0.356** | 0.206 | 0.358 | 0.264 | 0.413 | 0.223 | 0.371 | 0.214 | 0.359 | 0.329 | 0.445 | 0.335 | 0.467 | 0.418 | 0.523 | 1.016 | 0.877 |
| ETTm1 | **0.052** | **0.172** | 0.053 | 0.172 | 0.059 | 0.186 | 0.080 | 0.218 | 0.070 | 0.196 | 0.098 | 0.228 | 0.177 | 0.343 | 0.165 | 0.334 | 0.695 | 0.704 |
| ETTm2 | 0.125 | 0.263 | **0.121** | **0.258** | 0.194 | 0.345 | 0.174 | 0.323 | 0.129 | 0.266 | 0.256 | 0.362 | 0.222 | 0.377 | 0.232 | 0.384 | 0.316 | 0.456 |
| Electricity | 0.406 | 0.470 | 0.395 | 0.458 | 1.047 | 0.826 | 0.619 | 0.587 | 0.383 | 0.446 | 0.331 | 0.411 | 0.930 | 0.780 | 0.872 | 0.747 | 0.795 | 0.715 |
| Traffic | 0.325 | 0.417 | 0.328 | 0.420 | 1.840 | 1.165 | 0.315 | 0.413 | 0.268 | 0.356 | **0.156** | **0.249** | 1.573 | 1.017 | 0.701 | 0.617 | 1.133 | 0.849 |
| Weather | **0.002** | **0.030** | 0.002 | 0.031 | **0.002** | **0.030** | 0.008 | 0.068 | 0.006 | 0.061 | 0.002 | 0.031 | 0.006 | 0.063 | 0.007 | 0.067 | 0.004 | 0.061 |
| Illness | 1.363 | 0.982 | 1.636 | 1.092 | 1.198 | 0.924 | 1.063 | 0.886 | **0.645** | **0.672** | 0.674 | 0.644 | 8.437 | 2.647 | 9.022 | 2.782 | 7.949 | 2.588 |

Crucially, these results demonstrate the complementary nature of the numerical and textual representations. For instance, on the ETTh1 dataset, NumText reduces the MSE by 2.5% compared to the stronger Num Only baseline. This confirms our central hypothesis that the textual expert provides valuable contextual information that the numerical-only model cannot capture, leading to a synergistic improvement in performance.

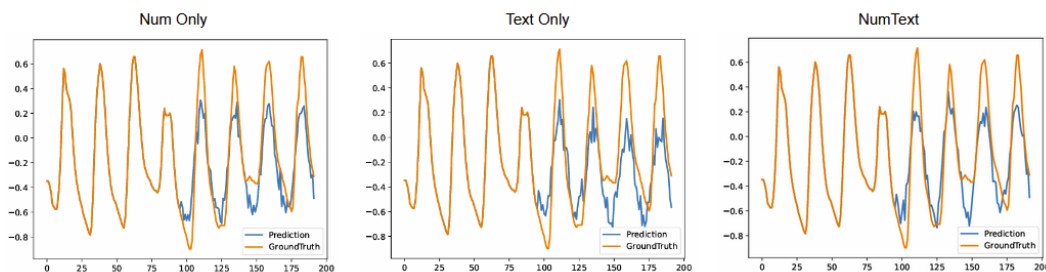

Figure 4: A qualitative case study on the ETTh2 dataset providing visual evidence for our hypothesis that numerical and textual modalities are complementary. More case studies can be found in A.5.

Figure 4 provides visual support for our hypothesis that numerical and textual representations are complementary. The figure presents a case study on the ETTh2 dataset, comparing our full **NumText** model against its unimodal baselines.

The **Num Only** and **Text Only** models, when used in isolation, exhibit distinct drawbacks. Their predictions appear noisy and struggle to precisely align with the ground truth's peaks and dips, suggesting that each modality alone provides a flawed view of the underlying patterns.

In contrast, the **NumText** forecast is visibly superior, producing a smoother and more stable prediction that accurately tracks the ground truth. This illustrates the synergy of the two modalities: one expert likely provides the robust underlying structure, while the other refines the fit to capture high-frequency details, effectively canceling out the noise and imprecision present in the unimodal forecasts. This successful fusion demonstrates that the numerical and textual views are indeed complementary, and that combining them allows the model to generate more accurate and reliable results.

To avoid MoE collapse where the gating network learns to route a vast majority of inputs only to a single expert, we checked the output weights of our gating network using a PyTorch forward hook during inference. Our analysis confirmed that the model consistently utilized both experts, with the average importance weights showing a near 50-50 split between the two modalities, indicating that no collapse had occurred and that both are considered equally important.

## 5.4 COMPARISON WITH STATE-OF-THE-ART MODELS

As shown in Table 3, NumText achieves new state-of-the-art results on four of the eight benchmark datasets, proving particularly effective on the ETT datasets with their complex seasonal patterns. However, the success of other models on different data structures—such as DLinear on the lin-

ear trends of the Illness dataset Zeng et al. (2023) and iTransformer on the Traffic and Electricity datasets—underscores the critical role of the input representation.

This reinforces our central hypothesis that representation is key to performance. While our patching approach is effective, future work could integrate complementary techniques, like DLinear's trend-seasonality decomposition or iTransformer's inverted embeddings, to create an even more robust model.

### 5.5 PERFORMANCE WITH DIFFERENT LLM BACKBONES

To verify that our performance gains stem from our dual-modality architecture and not just the choice of a specific LLM, we tested our framework with three different backbones: Llama-3.2-1B-Instruct Dubey et al. (2024), Llama-3.2-3B-Instruct Dubey et al. (2024), and GPT2 Radford et al. (2019). As shown in Figure 5, while more powerful LLMs yield slightly better results, the performance difference between them is small. Crucially, the substantial gain of **NumText** over the **Num Only** baseline remains consistent across all backbones. This demonstrates that our proposed architecture provides a robust benefit that is not solely dependent on the underlying language model.

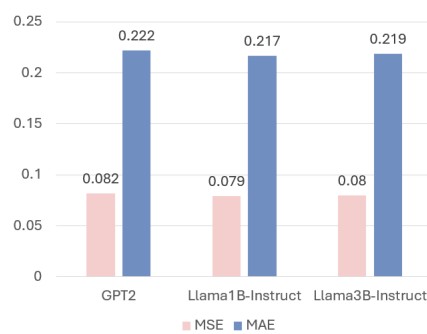

Figure 5: Ablation study of the performance gain from our dual-modality approach across three different LLM backbones on the ETTh1 dataset. The results show that the performance improvement is robust and consistent, regardless of the specific LLM used.

## 6 CONCLUSION

In this paper, we addressed a fundamental challenge in applying Large Language Models (LLMs) to time series forecasting: their inherent difficulty in processing raw numerical data. We argued that existing methods, which rely on elaborate prompts with hand-engineered features, are not truly end-to-end and risk data leakage. To overcome this, we introduced **NumText**, a novel dual-modality framework that learns a synergistic representation from both a direct numerical view and a textual view of the same raw time series data.

Our comprehensive experiments validate this approach. The ablation studies demonstrated that the fused representation is superior to either the numerical-only or textual-only modalities, confirming their complementary nature. By effectively combining the precise magnitude information from the numerical expert with the rich, contextual patterns captured by the LLM-based textual expert, NumText achieves new state-of-the-art results on four of the eight benchmark datasets. However, the success of other models on different data structures—such as **DLinear** on the linear trends of the **Illness** dataset and **iTransformer** on the **Traffic** and **Electricity** datasets—underscores the critical role of the input representation. This observation reinforces our central hypothesis that representation is key to performance, while also highlighting that our model's performance is robust across different LLM backbones.

For future work, these insights suggest several exciting avenues. An immediate step is to integrate complementary pre-processing techniques, like **DLinear's** trend-seasonality decomposition or **iTransformer's** inverted embeddings, to create an even more robust representation. Beyond representation, exploring more sophisticated fusion mechanisms, such as cross-attention, could yield further improvements over the current Mixture-of-Experts. Ultimately, it is worth noting that the performance boost is still modest given the vast amount of training data and parameters inherent to the LLM backbones. This suggests a potential ceiling imposed by current models' limited capacity for numerical processing, pointing to a critical need for a new generation of LLMs with fundamentally stronger, innate numerical understanding.

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

# A APPENDIX

## A.1 PROOF OF THEOREM 4.1

*Proof.* By the law of total variance,

$$\mathcal{R}(X) = \mathbb{E}\big[\,\mathrm{Var}(Y \mid X)\big] = \mathrm{Var}(Y) - \mathrm{Var}\big(\mathbb{E}[Y \mid X]\big).$$

Since $\sigma(E_{\mathrm{num}}) \subset \sigma(E_{\mathrm{num}}, E_{\mathrm{text}})$, conditioning on the joint variables increases variance of the conditional mean, thus reducing the residual risk. Equality holds iff $E_{\mathrm{text}}$ carries no extra information about $Y$ beyond $E_{\mathrm{num}}$. $\square$

## A.2 PROOF OF THEOREM 4.2

*Proof.* On each mixture component $p_k$, the conditional MSE is a strictly convex quadratic in $w$, minimized at $w_k^*$. Using the same static $w$ for all components implies at least one component suffers suboptimality. Summing over mixture weights $\pi_k$ yields a strictly larger total risk than the oracle dynamic strategy. $\square$

## A.3 PROOF OF THEOREM 4.3

*Proof.* For each $x$, $\mathrm{MSE}(x; w)$ is a strictly convex quadratic in $w$ with a unique minimizer $w^*(x)$. If $\Sigma(x)$ is not constant, then no static $w$ coincides with $w^*(x)$ everywhere, so on a set of positive measure the risk is strictly larger. Integrating over $x$ yields the result. $\square$

## A.4   NUMTEXT VS NUMONLY VS TEXTONLY

Table 4: Detailed comparison of model performance (MSE and MAE) across different modalities and prediction windows. **Bold** indicates the best performance for each metric on a given row.

| Data | Pred Window | NumText | | Num Only | | Text Only | |
|------|-------------|---------|---------|----------|----------|-----------|----------|
| | | MSE | MAE | MSE | MAE | MSE | MAE |
| ETTh1 | 96 | **0.057** | **0.182** | 0.058 | 0.183 | 0.060 | 0.190 |
| | 192 | **0.076** | **0.211** | 0.077 | 0.213 | 0.081 | 0.224 |
| | 336 | **0.090** | **0.236** | 0.092 | 0.238 | 0.094 | 0.244 |
| | 720 | **0.091** | **0.239** | 0.098 | 0.246 | 0.092 | 0.242 |
| ETTh2 | 96 | **0.136** | **0.284** | 0.139 | 0.288 | 0.150 | 0.301 |
| | 192 | **0.187** | **0.339** | 0.189 | 0.341 | 0.201 | 0.354 |
| | 336 | **0.231** | **0.385** | 0.232 | 0.385 | 0.235 | 0.390 |
| | 720 | 0.262 | 0.414 | 0.265 | 0.417 | **0.256** | **0.408** |
| ETTm1 | 96 | **0.029** | **0.126** | 0.029 | 0.126 | 0.029 | 0.126 |
| | 192 | **0.043** | **0.157** | 0.043 | 0.158 | 0.043 | 0.158 |
| | 336 | **0.056** | **0.182** | 0.057 | 0.184 | 0.057 | 0.184 |
| | 720 | **0.080** | **0.221** | 0.081 | 0.218 | 0.081 | 0.222 |
| ETTm2 | 96 | **0.069** | **0.192** | 0.070 | 0.192 | 0.070 | 0.195 |
| | 192 | 0.103 | 0.240 | **0.102** | **0.238** | 0.105 | 0.242 |
| | 336 | 0.135 | 0.280 | **0.130** | **0.274** | 0.140 | 0.286 |
| | 720 | 0.193 | 0.339 | **0.180** | **0.329** | 0.196 | 0.343 |
| Electricity | 96 | 0.377 | 0.455 | **0.354** | **0.435** | 0.401 | 0.469 |
| | 192 | 0.380 | 0.451 | **0.362** | **0.435** | 0.393 | 0.460 |
| | 336 | 0.413 | 0.470 | **0.405** | **0.461** | 0.433 | 0.483 |
| | 720 | **0.454** | **0.505** | 0.457 | 0.500 | 0.480 | 0.522 |
| Traffic | 96 | **0.326** | **0.417** | 0.343 | 0.430 | 0.338 | 0.428 |
| | 192 | **0.316** | **0.410** | 0.320 | 0.414 | 0.321 | 0.414 |
| | 336 | 0.316 | 0.413 | **0.313** | **0.410** | 0.324 | 0.419 |
| | 720 | 0.340 | 0.428 | **0.336** | **0.427** | 0.354 | 0.439 |
| Weather | 96 | **0.001** | **0.026** | 0.001 | 0.027 | 0.001 | 0.027 |
| | 192 | **0.002** | **0.029** | 0.002 | 0.030 | 0.002 | 0.029 |
| | 336 | **0.002** | **0.030** | 0.002 | 0.031 | 0.002 | 0.031 |
| | 720 | **0.002** | **0.034** | 0.002 | 0.036 | 0.002 | 0.034 |
| Illness | 24 | **1.113** | **0.877** | 1.551 | 1.057 | 1.331 | 0.983 |
| | 36 | **1.258** | **0.940** | 1.542 | 1.056 | 1.413 | 1.019 |
| | 48 | **1.419** | **1.009** | 1.622 | 1.089 | 1.510 | 1.060 |
| | 60 | **1.663** | **1.103** | 1.830 | 1.167 | 1.744 | 1.148 |

## A.5 NUMTEXT VS NUMONLY VS TEXTONLY VISUALIZATION

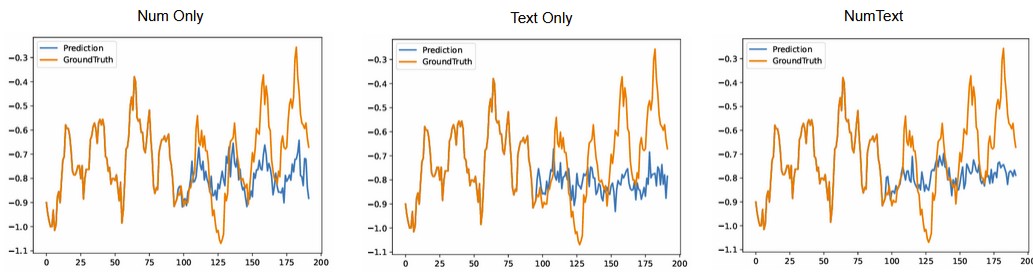

Figure 6: ETTh1 pred vs ground truth case study

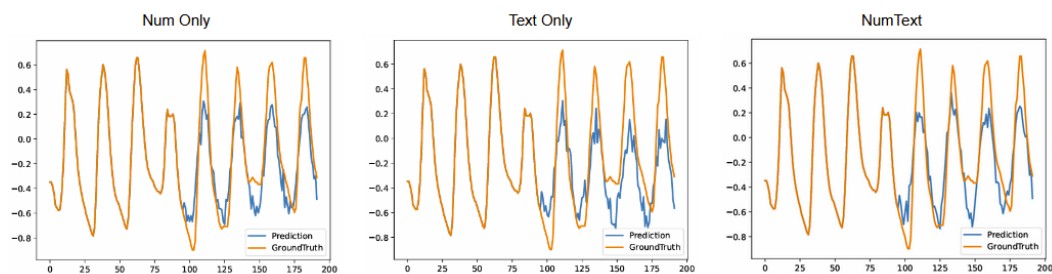

Figure 7: ETTh2 pred vs ground truth case study

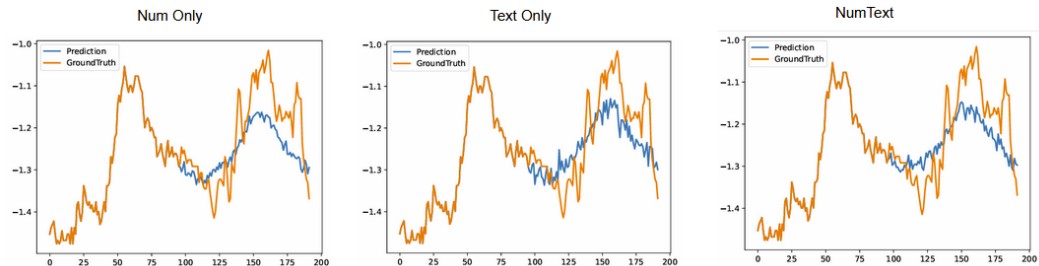

Figure 8: ETTm1 pred vs ground truth case study

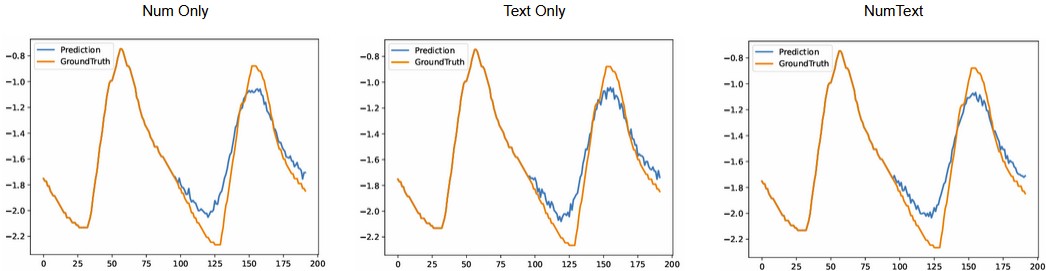

Figure 9: ETTm2 pred vs ground truth case study

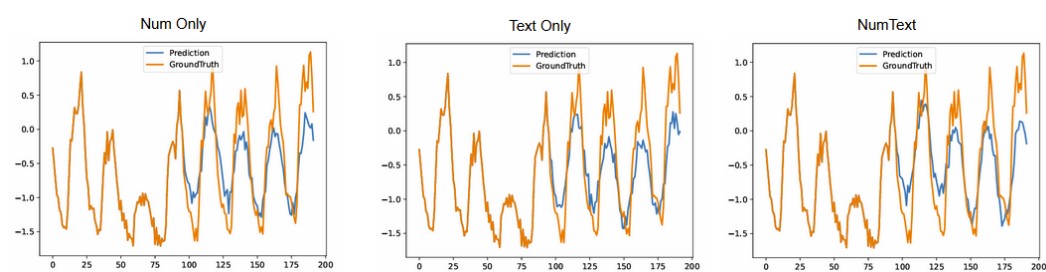

Figure 10: electricity pred vs ground truth case study

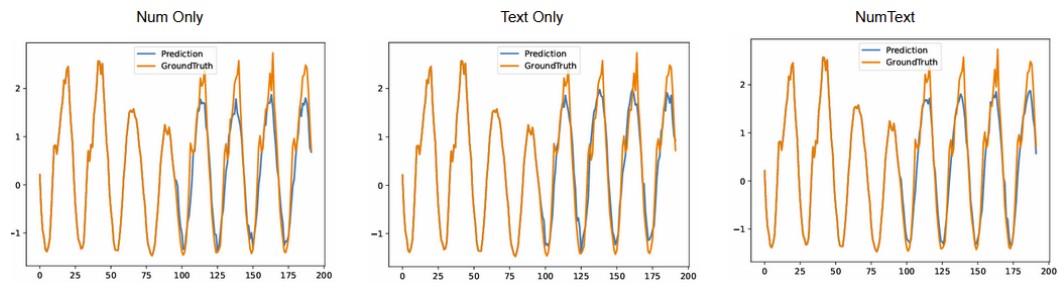

Figure 11: traffic pred vs ground truth case study

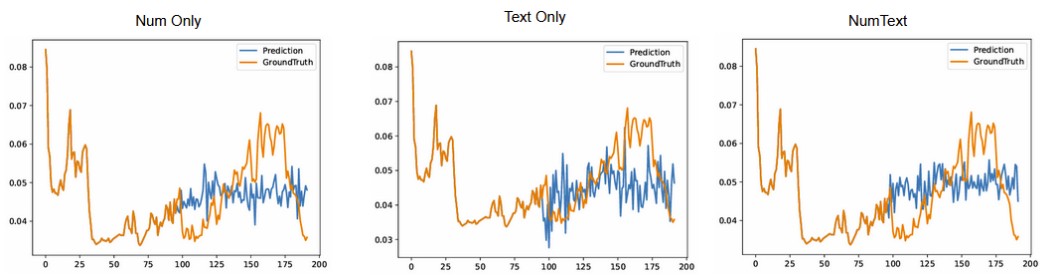

Figure 12: weather pred vs ground truth case study

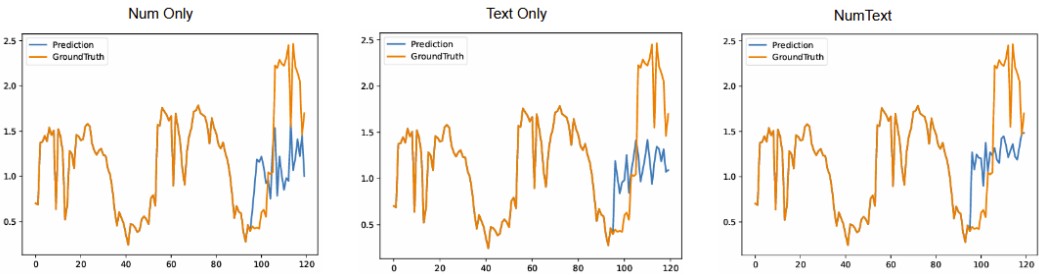

Figure 13: Illness pred vs ground truth case study

## A.6 COMPARISON WITH STATE-OF-THE-ART MODELS

Table 5: Detailed comparison of model performance (MSE and MAE) across different benchmarks and prediction windows. **Bold** indicates the green-marked cells from the source spreadsheet.

| Data | Pred Window | NumText MSE | NumText MAE | PatchTST MSE | PatchTST MAE | TimeCMA MSE | TimeCMA MAE | Autoformer MSE | Autoformer MAE | Dlinear MSE | Dlinear MAE | iTransformer MSE | iTransformer MAE | Crossformer MSE | Crossformer MAE | Pyraformer MSE | Pyraformer MAE | Reformer MSE | Reformer MAE |
|---|---|---|---|---|---|---|---|---|---|---|---|---|---|---|---|---|---|---|---|
| ETTh1 | 96 | **0.057** | **0.182** | 0.058 | 0.183 | 0.069 | 0.201 | 0.081 | 0.221 | 0.061 | 0.184 | 0.111 | 0.25 | 0.261 | 0.437 | 0.315 | 0.485 | 0.697 | 0.747 |
| | 192 | **0.076** | **0.211** | 0.077 | 0.213 | 0.084 | 0.225 | 0.093 | 0.235 | 0.082 | 0.214 | 0.13 | 0.277 | 0.284 | 0.455 | 0.343 | 0.502 | 1.079 | 0.942 |
| | 336 | **0.09** | **0.236** | 0.092 | 0.238 | 0.101 | 0.253 | 0.112 | 0.263 | 0.113 | 0.263 | 0.136 | 0.287 | 0.287 | 0.46 | 0.41 | 0.558 | 1.779 | 1.293 |
| | 720 | **0.091** | **0.239** | 0.098 | 0.246 | 0.107 | 0.258 | 0.122 | 0.278 | 0.217 | 0.39 | 0.168 | 0.324 | 0.413 | 0.569 | 0.519 | 0.642 | 1.772 | 1.295 |
| ETTh2 | 96 | 0.136 | 0.284 | 0.139 | 0.288 | 0.21 | 0.364 | 0.147 | 0.297 | **0.129** | **0.272** | 0.278 | 0.397 | 0.292 | 0.435 | 0.359 | 0.483 | 0.587 | 0.643 |
| | 192 | 0.187 | 0.339 | 0.189 | 0.341 | 0.251 | 0.402 | 0.189 | 0.341 | **0.18** | **0.326** | 0.32 | 0.438 | 0.316 | 0.453 | 0.387 | 0.503 | 0.815 | 0.772 |
| | 336 | **0.231** | **0.385** | 0.232 | 0.385 | 0.284 | 0.433 | 0.247 | 0.395 | 0.232 | 0.379 | 0.35 | 0.463 | 0.325 | 0.46 | 0.429 | 0.531 | 1.269 | 1.012 |
| | 720 | **0.262** | **0.414** | 0.265 | 0.417 | 0.31 | 0.453 | 0.31 | 0.45 | 0.315 | 0.458 | 0.368 | 0.48 | 0.406 | 0.52 | 0.495 | 0.576 | 1.394 | 1.079 |
| ETTm1 | 96 | **0.029** | **0.126** | **0.029** | **0.126** | 0.038 | 0.15 | 0.049 | 0.174 | 0.041 | 0.151 | 0.046 | 0.153 | 0.083 | 0.23 | 0.079 | 0.225 | 0.286 | 0.467 |
| | 192 | **0.043** | **0.157** | 0.043 | 0.158 | 0.051 | 0.174 | 0.067 | 0.206 | 0.061 | 0.185 | 0.077 | 0.205 | 0.143 | 0.316 | 0.132 | 0.3 | 0.444 | 0.601 |
| | 336 | **0.056** | **0.182** | 0.057 | 0.184 | 0.062 | 0.194 | 0.095 | 0.244 | 0.067 | 0.194 | 0.117 | 0.256 | 0.2 | 0.384 | 0.188 | 0.368 | 0.469 | 0.613 |
| | 720 | **0.08** | **0.221** | 0.081 | 0.218 | 0.083 | 0.224 | 0.107 | 0.248 | 0.112 | 0.252 | 0.153 | 0.297 | 0.283 | 0.465 | 0.26 | 0.441 | 1.58 | 1.134 |
| ETTm2 | 96 | **0.069** | **0.192** | 0.07 | 0.192 | 0.156 | 0.309 | 0.147 | 0.3 | 0.07 | 0.191 | 0.121 | 0.238 | 0.151 | 0.309 | 0.169 | 0.325 | 0.264 | 0.419 |
| | 192 | 0.103 | 0.24 | **0.102** | **0.238** | 0.178 | 0.328 | 0.209 | 0.35 | 0.109 | 0.246 | 0.198 | 0.32 | 0.19 | 0.35 | 0.204 | 0.361 | 0.313 | 0.455 |
| | 336 | 0.135 | 0.28 | **0.13** | **0.274** | 0.201 | 0.351 | 0.152 | 0.303 | 0.15 | 0.297 | 0.308 | 0.408 | 0.244 | 0.401 | 0.254 | 0.407 | 0.384 | 0.505 |
| | 720 | 0.193 | 0.339 | **0.18** | **0.329** | 0.241 | 0.39 | 0.188 | 0.339 | 0.187 | 0.331 | 0.397 | 0.482 | 0.303 | 0.447 | 0.299 | 0.442 | 0.302 | 0.444 |
| Electricity | 96 | 0.377 | 0.455 | 0.354 | 0.435 | 0.986 | 0.801 | 0.429 | 0.491 | 0.373 | 0.437 | **0.252** | **0.362** | 0.863 | 0.753 | 0.857 | 0.745 | 0.721 | 0.676 |
| | 192 | 0.38 | 0.451 | 0.362 | 0.435 | 1.029 | 0.82 | 0.543 | 0.556 | 0.351 | 0.422 | **0.296** | **0.388** | 0.912 | 0.775 | 0.817 | 0.72 | 0.751 | 0.686 |
| | 336 | 0.413 | 0.47 | 0.405 | 0.461 | 1.048 | 0.825 | 0.764 | 0.672 | 0.381 | 0.443 | **0.356** | **0.424** | 0.956 | 0.788 | 0.879 | 0.746 | 0.824 | 0.728 |
| | 720 | 0.454 | 0.505 | 0.457 | 0.5 | 1.124 | 0.858 | 0.741 | 0.628 | 0.426 | 0.483 | **0.421** | **0.471** | 0.989 | 0.805 | 0.935 | 0.775 | 0.882 | 0.769 |
| Traffic | 96 | 0.326 | 0.417 | 0.343 | 0.43 | 1.845 | 1.167 | 0.327 | 0.434 | 0.299 | 0.395 | **0.146** | **0.234** | 0.578 | 0.551 | 0.69 | 0.614 | 1.059 | 0.814 |
| | 192 | 0.316 | 0.41 | 0.32 | 0.414 | 1.835 | 1.165 | 0.324 | 0.418 | 0.245 | 0.334 | **0.151** | **0.239** | 1.923 | 1.182 | 0.684 | 0.606 | 1.136 | 0.849 |
| | 336 | 0.316 | 0.413 | 0.313 | 0.41 | 1.851 | 1.168 | 0.28 | 0.385 | 0.241 | 0.33 | **0.152** | **0.246** | 1.834 | 1.145 | 0.698 | 0.615 | 1.174 | 0.869 |
| | 720 | 0.34 | 0.428 | 0.336 | 0.427 | 1.827 | 1.159 | 0.328 | 0.415 | 0.285 | 0.366 | **0.175** | **0.275** | 1.955 | 1.191 | 0.732 | 0.634 | 1.162 | 0.863 |
| Weather | 96 | **0.001** | **0.026** | 0.001 | 0.027 | 0.001 | 0.027 | 0.007 | 0.066 | 0.003 | 0.041 | 0.001 | 0.027 | 0.006 | 0.065 | 0.006 | 0.062 | 0.002 | 0.038 |
| | 192 | **0.002** | **0.029** | 0.002 | 0.03 | **0.002** | **0.029** | 0.012 | 0.089 | 0.006 | 0.065 | **0.002** | **0.029** | 0.006 | 0.063 | 0.006 | 0.067 | 0.005 | 0.054 |
| | 336 | **0.002** | **0.03** | 0.002 | 0.031 | **0.002** | **0.03** | 0.007 | 0.06 | 0.006 | 0.067 | 0.002 | 0.031 | 0.005 | 0.057 | 0.007 | 0.068 | 0.005 | 0.056 |
| | 720 | **0.002** | **0.034** | 0.002 | 0.036 | **0.002** | **0.034** | 0.005 | 0.058 | 0.007 | 0.07 | 0.002 | 0.035 | 0.007 | 0.067 | 0.007 | 0.07 | 0.005 | 0.057 |
| Illness | 24 | 1.113 | 0.877 | 1.551 | 1.057 | 1.276 | 0.95 | 0.96 | 0.821 | **0.437** | **0.538** | 0.627 | 0.577 | 8.403 | 2.653 | 8.756 | 2.73 | 8.037 | 2.604 |
| | 36 | 1.258 | 0.94 | 1.542 | 1.056 | 1.189 | 0.918 | 0.967 | 0.851 | **0.6** | **0.651** | 0.653 | 0.627 | 8.874 | 2.708 | 9.056 | 2.788 | 7.633 | 2.529 |
| | 48 | 1.419 | 1.009 | 1.622 | 1.089 | 1.118 | 0.888 | 1.048 | 0.891 | 0.709 | 0.715 | **0.685** | **0.663** | 8.013 | 2.585 | 9.075 | 2.788 | 8.06 | 2.616 |
| | 60 | 1.663 | 1.103 | 1.83 | 1.167 | 1.209 | 0.941 | 1.278 | 0.982 | 0.832 | 0.783 | **0.73** | **0.709** | 8.458 | 2.641 | 9.202 | 2.82 | 8.065 | 2.603 |

## DISCLOSURE ON THE USE OF LARGE LANGUAGE MODELS

During the preparation of this manuscript, the authors utilized a large language model (Google's Gemini) to assist with improving grammar, refining phrasing, and enhancing the clarity of the text. The core scientific contributions, including the proposed methodology, experimental design, and interpretation of results, are solely the work of the authors, who take full responsibility for the final content of this paper.

