# OpenReview forum: "NUMBERS AS TEXT: COMPLEMENTARY DUAL- MODALITY EMBEDDINGS FOR TIME SERIES FORECASTING"
_ICLR.cc/2026/Conference — Submitted to ICLR 2026_

### Official Review · Reviewer_751u · 2025-10-26

**Soundness:** 2
**Presentation:** 2
**Contribution:** 3
**Rating:** 4
**Confidence:** 5

**Summary:**

This paper presents a framework for time series forecasting. It treats the series as a dual-modality input, generating the numerical value embeddings and the LLM output embeddings, which are then combined through a modality-specific Mixture-of-Experts (MoE). Experiments reveal that the numerical embeddings and the textual embeddings are complementary benefits for time series forecasting.

**Strengths:**

This paper assumes that the time series as text is the most effective method for LLM-based time series forecasting. I like the new experimental settings and theoretical justification of this paper. The code is available, which increases the credibility of this paper.

**Weaknesses:**

The presented "Raw Data Input" and "Time Series as Numerical Value" have already been proposed in existing studies [1-3]. The dual-modality architecture and time-series text embedding cache for reducing computational costs have already been proposed in [3]. Please elaborate on the differences between the existing techniques in SOAT works and this paper in more detail.

[1]  Large language models are zero-shot time series forecasters[J]. Advances in Neural Information Processing Systems, 2023, 36: 19622-19635.

[2] Promptcast: A new prompt-based learning paradigm for time series forecasting[J]. IEEE Transactions on Knowledge and Data Engineering, 2023, 36(11): 6851-6864.

[3] Timecma: Towards llm-empowered multivariate time series forecasting via cross-modality alignment[C]//Proceedings of the AAAI Conference on Artificial Intelligence. 2025, 39(18): 18780-18788.

**Questions:**

- Could you please clarify your contribution and existing techniques in Weakness 1?

- In Theorem 4.1, the authors claimed that MSE using both modalities is never worse than using a single modality. May I ask the specific scenario? If modality in modality out, I agree with this Theorem. But if modality in single modality (time series) is out, is the Theorem still established? As external modality would be the noise for the single modality output if there are no effective alignment strategies. A recent study [4] also points out that "Multimodal time series forecasting models do not consistently outperform the strongest unimodal baselines."

- How about the time for generating and reading the cache? These costs would be considered.

[4] Does Multimodality Lead to Better Time Series Forecasting?[J]. arXiv preprint arXiv:2506.21611, 2025.

---

### Official Review · Reviewer_csUN · 2025-10-26

**Soundness:** 2
**Presentation:** 2
**Contribution:** 2
**Rating:** 2
**Confidence:** 5

**Summary:**

NumText presents a dual-modality representation for long-term TSF that fuses a direct numerical value embedding with a textual embedding obtained by quantizing raw segments into integer tokens and encoding them with a frozen LLM. A soft MoE performs input-dependent weighting of the two modalities, followed by a standard Transformer encoder and a linear forecasting head. The paper evaluated on 8 benchmarks and found that numerical and LLM-textual views are complementary, and that dynamic fusion outperforms static fusion.

**Strengths:**

1. A simple dual-path (Numerical vs Textual Expert) with a soft MoE and a standard Transformer head is easy to implement and analyze.
2. The textual input is mechanically derived via quantization from raw numbers; no dataset-specific statistics/metadata prompts, reducing lookahead bias risk.
3. Ablations (Num-only / Text-only / Dual) show consistent, supporting the hypothesis; plus a caching trick for LLM embeddings improves efficiency.

**Weaknesses:**

1. Missing critical LLM-TSF baselines results, such as GPT4TS, Time-LLM and TEMPO [1], making the evidence for superiority incomplete.
2. Limited novelty for “LLM-Based TSF”: The core novelty is a straightforward quantize-to-text + frozen LLM embedding + MoE fusion. This is not a fundamentally new LLM-TSF paradigm.
3. The performance gains over strong baselines are modest, which does not substantiate the claimed effectiveness.
4. The method uses a mean-of-patches as context; the paper lacks deeper analyses (e.g., per-regime routing behavior, sensitivity to quantization, or cross-attention structure vs MoE). This limits insight into when and why textual LLM embeddings help.

[1] TEMPO: Prompt-based Generative Pre-trained Transformer for Time Series Forecasting

**Questions:**

In addition to the commonly used 8 benchmarks, would you consider including more datasets, such as the Monash TSF Archive (e.g., M3/M4/M5), Exchange Rate, Solar, Wiki-traffic (non-stationary), retail demand (intermittent), and healthcare data (e.g., ICU vitals with missingness)?

---

### Official Review · Reviewer_sD2x · 2025-10-28

**Soundness:** 2
**Presentation:** 2
**Contribution:** 2
**Rating:** 2
**Confidence:** 5

**Summary:**

This paper introduces NumText, which is a dual-modality framework that integrates the numerical and textual representations of time series data to improve time series forecasting performance. A time series patch is processed by two experts, specifically a textual expert, which converts normalized numerical values into quantized text tokens and extracts semantic embeddings via a frozen LLM, and a numerical expert, which linearly projects the raw numeric patch into embedding space.  It also designs a soft Mixture-of-Experts module that dynamically fuses both modality representations. A Transformer backbone and projection head are used for forecasting.  Empirical results across eight benchmark datasets show that the NumText achieves good performance results and improves results over unimodal baselines.

**Strengths:**

1. I support that when using LLMs for time series forecasting, inputting numerical inputs as text is more effective than incorporating contextual descriptions ( this may work when classification or other decision boundary involved time series analysis task). The idea of treating raw TS data simultaneously as numbers and text is great.

2. Providing a theoretical proof about incorporating two different modalities is beneficial than unimodal forecasting and soft-MoE is effective.

3. Comprehensive experimental evaluation on benchmark datasets and compare with baseline models. Ablation study shows the effectiveness of modalities and minimal effects on different choices of LLM.

**Weaknesses:**

1. Components for time series process ( ReINV, Patch, text tokenization etc.) are widely used in current time series analysis pipelines. Thus, the technical novelty of this paper is limited.
2. Related work is not up-to-date and missing important baseline comparisons, such as time-LLM.
3. Your experimental results appear promising, particularly on the weather datasets. However, the visualizations suggest inconsistencies, which I doubt the results.

**Questions:**

I looked into your code, but didn't find the soft-MoE design. Since enc_out, n_vars = self.patch_embedding(x_enc) followed with enc_out, attns = self.encoder(enc_out). It would be great if you could point the soft-MoE code to me.

---

### Official Review · Reviewer_TNs5 · 2025-10-29

**Soundness:** 2
**Presentation:** 3
**Contribution:** 1
**Rating:** 2
**Confidence:** 5

**Summary:**

This paper studies time series forecasting (TSF) by introducing NumText, a dual-modality framework to extract embeddings from (1) the raw time series, and (2) the textual prompt (which includes the time series), and then use a MoE module to fuse the embeddings of (1) and (2) for further forecasting. Extensive experiments have been conducted to verify the effectiveness of each component, validating the complementarity of numerical and textual representations.

**Strengths:**

1. The core idea of separately extracting embeddings from two modalities and fuse them for TSF is reasonable.
2. Some theoretical justifications are included to verify the superiority of dynamic MoE over single-modality or static fusion methods.
3. Experiments have been conducted with different LLMs to demostrate the robustness of the proposed framework.

**Weaknesses:**

1. In L45-L49, the description of 2 problems are farfetched:
- I disagree (1) is a fundamental problem, existing methods just use some simple statistics.
- In (2), LLMs have not been trained with time series corpus, I don't think future data would appear during training.
- There are no clues to prove the existence of (2) and no experiments have been conducted to validate that this paper can prevent from (2).
2. In L80-L81, it's unclear why representing time series as text can directly unlock LLM's forecasting ability - there should theoretically be a clear gap between sentences and time series.
3. In L113-L115, the wordings like "first to systematically ..." is overclaimed - in the baseline TimeCMA (mentioned in L109), it has already studied such approach.
4. It seems this method just replaces the cross attention of TimeCMA, and shorten the textual prompt, please discuss how this paper differs to TimeCMA.
- The techinical contribution of this paper is quite limited.
- Besides, there are no experiments to demonstrate the superiority of MoE (this paper) over the cross attention (TimeCMA).
5. In Table 2, the performance of three settings are quite similar in almost all cases, and the incorporation of LLMs would inevitably make the inference speed much slower, making it unnecessary to introduce LLM for TSF.
6. In Table 3:
- Please include more recent and stronger advances for comparisons, e.g., TimeXer.
- Please include more LLM-based methods for comaprisons, e.g., Time-LLM.
- The results of TimeCMA look quite strange - worse than PatchTST in most cases.

**Questions:**

N.A.

---

### Meta-Review · Area_Chair_UB24 · 2026-01-03

**Summary:**

This paper tries to provide a dual-input paradigm for time series forecasting, where both numerical embedding and LLM-based embedding are encoded via linear projection and LLM respectively for the forecasters.

**Reviewer Concerns:**

Reviewers hold crucial concerns regarding the quality and technical contributions of this paper. The studied idea has been extensively explored in the literature, especially in the area of LLM for time series forecasting. In addition, the proposed solution is a naive combination of existing methods, such as RevIN, quantization, de-normalization, linear projection, and LLM-based embedding. Worse still, from the submitted code the reviewer cannot find the implementation of Soft-MoE, which questions the validity of the whole solution.

**Reviewer Scores:**

Authors did not submit a rebuttal. The reviews are highly consistent in the negative side of overall recommendation. Their scores and opinions would not change even if rebuttal were provide, because the original submission falls short in many aspects that hinders further consideration.

---

### Decision · Program_Chairs · 2026-01-26

Reject